# High Incidence of Thyroid Cancer in Southern Tuscany (Grosseto Province, Italy): Potential Role of Environmental Heavy Metal Pollution

**DOI:** 10.3390/biomedicines11020298

**Published:** 2023-01-20

**Authors:** Marco Capezzone, Massimo Tosti Balducci, Eugenia Maria Morabito, Cosimo Durante, Paolo Piacentini, Liborio Torregrossa, Gabriele Materazzi, Giacomo Giubbolini, Virginia Mancini, Maja Rossi, Massimo Alessandri, Alessandra Cartocci

**Affiliations:** 1UOSD of Endocrinology, Misericordia Hospital, 58100 Grosseto, Italy; 2Unit of Nuclear Medicine, Misericordia Hospital, 58100 Grosseto, Italy; 3Department of Translational and Precision Medicine, Sapienza University of Rome, 00184 Rome, Italy; 4Unit of Epidemiology, Department of Prevention, Misericordia Hospital, 58100 Grosseto, Italy; 5Department of Surgical, Medical and Molecular Pathology, University Hospital of Pisa, 56124 Pisa, Italy; 6Division of Endocrine Surgery, Department of Surgical Pathology, University Hospital of Pisa, 56124 Pisa, Italy; 7Department of Pathology, Misericordia Hospital, 58100 Grosseto, Italy; 8Section of Pathology, Department of Medical Biotechnology, University of Siena, 53100 Siena, Italy; 9Laboratory Medicine Functional Area, Hospital Misericordia, 58100 Grosseto, Italy; 10Department of Medical Biotechnologies, Bioengineering Lab, University of Siena, 53100 Siena, Italy

**Keywords:** thyroid cancer, incidence, epidemiology, heavy metals

## Abstract

The incidence of thyroid cancer (TC) in Italy is one of the highest in Europe, and the reason for this is unclear. The intra-country heterogeneity of TC incidence suggests the possibility of an overdiagnosis phenomenon, although environmental factors cannot be excluded. The aim of our study is to evaluate the TC incidence trend in southern Tuscany, Italy, an area with particular geological characteristics, where the pollution and subsequent deterioration of various environmental matrices with potentially toxic elements (heavy metals) introduced from either geological or anthropogenic (human activities) sources are documented. The Tuscany cancer registry (ISPRO) provided us with the number of cases and EU standardized incidence rates (IR) of TC patients for all three provinces of southeast Tuscany (Siena, Grosseto, Arezzo) during the period of 2013–2016. In addition, we examined the histological records of 226 TC patients. We observed that the TC incidence rates for both sexes observed in Grosseto Province were significantly higher than those observed in the other two provinces. The increase was mostly due to the papillary (PTC) histotype (92% of cases), which presented aggressive variants in 37% of PTCs and tumor diameters more than 1 cm in 71.3% of cases. We demonstrated a high incidence of TC in Grosseto province, especially among male patients, that could be influenced by the presence of environmental heavy metal pollution.

## 1. Introduction

Thyroid cancer (TC) is the most common endocrine malignancy, but it is relatively rare (∼5% of all thyroid nodules) despite the high prevalence of thyroid nodules in the general population. A history of rapid nodular growth, fixation of the nodule to surrounding tissues, new-onset hoarseness or vocal cord paralysis, or the presence of ipsilateral cervical lymphadenopathy are factors that raise the suspicion that a nodule may be malignant. Although thyroid cancers are most common in euthyroid patients, in some cases, they could be associated with hyperthyroidism or hypothyroidism symptoms. It is known that a higher serum TSH level is associated with an increased risk of malignancy in a thyroid nodule, as well as more advanced-stage thyroid cancer [1,2,3].

The most common form of TC, accounting for more than 90% of cases, is differentiated thyroid cancer (DTC), originating from thyroid follicular epithelial cells, which include papillary (PTC), follicular (FTC), and oncocytic cell thyroid cancers. On the contrary, poorly differentiated thyroid cancer and anaplastic thyroid cancer (ATC) are rare but more aggressive follicular-derived thyroid cancers. Medullary thyroid cancer originates in the parafollicular neuroendocrine cells of the thyroid and is uncommon, accounting for 1–2% of all thyroid cancers. Rare neoplasms of other origin include thyroid lymphomas (usually B-cell) and thyroid sarcomas. The most well-established risk factor for TC is exposure to ionizing radiation of the head and neck region in childhood, while other possible risk factors, such as chromosomal and genetic alterations, iodine intake, TSH level, autoimmune thyroid disease, gender, estrogen, obesity, lifestyle changes, and environmental pollutants, have also been postulated [4]. Most patients with thyroid cancer have an excellent prognosis with a high rate of disease-free survival. Risk stratification provides important information about an individual patient’s risk of recurrence and disease-specific mortality. It is usually performed using the Eighth Edition of the American Joint Committee on Cancer/tumor node metastasis (AJCC/TNM) staging system to predict disease-specific mortality and with the American Thyroid Association (ATA) Risk Stratification System to predict the risk of recurrent or persistent disease. These initial risk estimates are then modified over time using the descriptions from the ATA guidelines to define the patient’s response to therapy at any point during follow-up as excellent (no evidence of persistent/recurrent disease), biochemically incomplete (abnormal thyroglobulin (Tg) or rising Tg antibodies in the absence of identifiable structural disease), structurally incomplete (structural evidence of persistent/recurrent disease), or indeterminate (nonspecific findings that cannot be confidently classified as benign or malignant) [5,6]. These modified risk estimates are then used to plan ongoing management. The mainstay of treatment for differentiated TC includes a combination of surgery, radioactive iodine (RAI), and levothyroxine suppression. Radioactive iodine refractory differentiated thyroid cancer has a lower survival rate, prompting the use of other therapeutic options available. Over the past decade, the identification of genetic mutations in the signaling pathway involved in thyroid tumorigenesis has led to the approval of the tyrosine kinase inhibitors (TKIs) Sorafenib and Lenvatinib in RAI-refractory DTC. Similarly, metastatic medullary thyroid cancer (MTC) implies an unfavorable 10-year survival rate of only 20%, as the principal treatment options focus on locoregional control via surgical and/or non-surgical options. The approval of TKIs such as Cabozantinib and Vandetanib and highly selective RET inhibitors (Pralsetinib and Selpercatinib) has introduced an encouraging, novel, and systemic therapeutic option for metastatic MTC. Lastly, anaplastic thyroid cancer (ATC) carries the worst prognosis with high recurrence rates. Treatment includes surgery, chemotherapy, and external beam radiation. The FDA recently approved Dabrafenib plus trametinib for BRAF V600E mutated ATC [7,8].

The global incidence of TC has increased considerably during the past three decades across different populations [9]. The Italian TC incidence is one of the highest in Europe, although with intra-country heterogeneity, and according to Globocan 2020 estimates, it ranks second after Cyprus [10].

Many studies have attributed the rise of TC incidence to an overdiagnosis phenomenon resulting from the detection and diagnosis of small indolent tumors that would not otherwise have caused symptoms or death during an individual’s lifetime [11,12,13]. However, a growing number of observations have suggested potential environmental and lifestyle-related factors affecting TC incidence [14,15,16]. These hypotheses seem reinforced by recent evidence that shows a decline in TC incidence in the United States since 2014 and that similar declines are noted for PTC, localized disease, and microcarcinomas but not for tumors >2.0 cm [17]. In addition, mortality rates continue to increase [18,19]. The explanation for this ever-increasing number of TC diagnoses is probably multifactorial, with environmental factors also contributing to this TC incidence trend [20].

In Italy, there is currently no single national register of cancer cases, but rather complete and satisfactory regional registers of cancer (CR) [21]. A recent study reported that, in Italy, among TC cases diagnosed in the period from 1998 to 2012, the incidence rate was largely influenced by overdiagnosis: 75% of cases in women and 63% in men were overdiagnosed [22]. The wide heterogeneity of TC incidence across Italian areas suggests a possible role of medical practices, but the exposure to environmental factors cannot be completely excluded. For example, in the Sicily region, some studies showed important regional differences in the incidence of TC that was significantly higher in the volcanic area of Mt. Etna (northeastern Sicily) compared to the rest of the region, suggesting a possible relationship between the complex mixture of pollutants and TC diagnoses [23]. Southern Tuscany (Grosseto Province, Italy) is an area with geological peculiarities that has a long tradition of mining. In the Tuscany region, there are two important areas where mining activities are concentrated: the Monte Amiata and Metal Hills Districts located in Grosseto Province. Specifically, the Tuscan provinces with the highest number of metal mining sites are Grosseto (*n* = 41), Siena (*n* = 12), Livorno (*n* = 12), Pisa (*n* = 11), Firenze (*n* = 2), and Massa Carrara (*n* = 1). There are metal mining sites in Arezzo, Lucca, Prato, and Pistoia [24].

The aim of our study is to describe the thyroid cancer incidence trend in southern Tuscany, focusing on differences by gender, age at diagnosis, tumor size, TNM, histotype, and histological variants.

## 2. Materials and Methods

### 2.1. Study Population

The study was conducted in Grosseto Province, Tuscany Region, central Italy. This province is 4503 km^2^ large and has a population of around 217,000. The province is administratively divided into 28 municipalities belonging to five regional zones: Amiata Mountain, Metal Hills District, Albegna Hills, Orbetello Lacuna, and Grosseto area. Municipalities represent the basic administrative units in Italy. Health districts are intermediate units that coordinate health care delivery within the national health system. Grosseto Province belongs to the local health authority (ASL Southeast Tuscany), divided into eight health districts. A Cancer Registry has been active for the whole Tuscany region since 2013. The Tuscan Cancer Registry is certified by the Italian Association of Cancer Registries (AIRTUM), which performs validation checks on completeness of coverage, accuracy, and interpretation to assure standard quality (accessed on 1 March 2022, http://www.registri-tumori.it/cms/it/Accreditamento). The Tuscany Cancer Registry provided us with the number of cases and European age-standardized incidence rate (EU standardized IR) of thyroid cancer patients for all three provinces of southern Tuscany during the period from 2013 to 2016. In addition, we examined the histological records of TC patients diagnosed in 2013–2016. The large majority of TC patients were born in Grosseto or were residents for at least 20 years. All histologic slides were reexamined and reclassified according to the pTNM 8th edition criteria by an endocrine pathologist (L.T.). We did not distinguish between histological types of thyroid cancer; however, the large majority were papillary thyroid cancers (PTCs). Clinical (age at diagnosis, sex, municipality of residence, type of surgery) and pathological data (tumor size, distant/local metastases, extrathyroidal extension, multifocality, and bilaterality of the tumor) were collected and recorded in a database. In this study, it was not possible to evaluate the clinical outcome of the TC patients because many of them were followed in other Italian hospitals.

### 2.2. Statistical Analysis

Descriptive statistics were carried out. Qualitative variables were described as absolute frequencies and percentages, while the quantitative ones were described as mean ± standard deviation (SD) or median and interquartile range if the variable was non-normally distributed. The chi-squared or Kruskal tests were used to evaluate the association or difference of TC population in Grosseto between the main five areas of the province. Crude incidence rate and/or EU-standardized IR and their 95% confidence intervals (95% CI) were estimated. The standardized incidence ratio (SIR) and its 95% CI were estimated to compare two incidence rates (i.e., if SIR and its confidence interval is <1, the reference incidence rate is statistically higher than the other one; instead, if SIR and its confidence interval is >1, the reference incidence rate is statistically lower than the other one).

## 3. Results

### 3.1. Thyroid Cancer Incidence in Southeast Tuscany

Comparing the incidence rates of TC between the three Asl provinces of southeast Tuscany from 1 January 2013 through 31 December 2016, we observed that the TC incidence rates for both sexes observed in Grosseto Province (among men, EU standardized IR = 17.66 per 100,000 residents per year, 95% CI = 12.46 to 19.89; among women, EU standardized IR = 34.16 per 100,000 residents per year, 95% CI = 28.36 to 39.43) were significantly higher than those observed in the other two provinces of southeast Tuscany (Table 1). In detail, Siena’s province had an EU-standardized IR for women of 25.30 per 100,000 (95% CI: 20.86–29.43), and comparing this incidence with Grosseto, a SIR of 0.75 (95% CI: 0.63–0.88) was obtained; among men, EU-standardized IR was 8.51 per 100,000 (95% CI 15.86–10.94), and the SIR was 0.49 (95% CI: 0.35–0.65). In addition, Siena’s IR was not significantly different from that observed in Italy, being 0.90 (95% CI: 0.76–1.06) for females and 0.84 (95% CI: 0.61–1.14) for males. In Italy, the thyroid cancer crude IR was 10.1 per 100,000 residents per year among males and 28.2 per 100,000 residents per year among females. As for Arezzo Province, an EU-standardized IR of 22.20 per 100,000 (95% CI: 19.05–26.36) for women was estimated, with a SIR of 0.66 (95% CI: 0.56–0.77) when compared to the Grosseto area; among men, the EU-standardized IR was 8.58 per 100,000 (95% CI: 6.48–10.88), and the SIR was 0.53 (95% CI: 0.40–0.68). Compared to the national data, the SIR was 0.79 (95% CI: 0.67–0.92) for females and 0.92 (95% CI: 0.70–1.18) for males.

The incidence was also estimated across the five zones in Grosseto to understand more deeply if some of them have higher incidence rates. Regarding the TC incidence in the female population, we can observe from Figure 1A that the incidence is higher in the Grosseto area (crude IR 30.6) and in Amiata Mountain (crude IR 28.7). From Figure 1B, the differences between these areas seem less evident, except for Grosseto; however, the crude incidence rate of all the zones ranged from 13.6 in the Metal Hills District to 15.8 in the Grosseto zone, and they are not statistically different. It should also be pointed out that the Italian incidence rate for the male population is 10.1, so the incidence for each zone is at least 35% higher than the national average.

### 3.2. Clinical and Pathological Features of TC Patients (n = 226) Living in Grosseto Province

We examined the histological records of 226/235 (96.2%) TC patients living in Grosseto Province. We did not have the histological data of nine patients because they were followed in other Italian regions. As shown in Table 2, most patients (208/226, 91.6%) were affected by papillary thyroid cancer (PTC), and the remaining 18 patients had a follicular histotype (FTC) in 8 cases (3.5%), medullary (MTC) in 7 cases (3.1%), and undifferentiated/anaplastic in 3 cases (1.8%). The PTC variants were reported in 176 cases: 48/176 (27.3%) were the classical variant, 63/176 (35.8%) follicular variants, 35/176 (19.9%) tall cell variants, 25/176 (14.2%) solid variants, and 5/176 (2.8%) diffuse sclerosing variants. The median age of patients was 52 years (range 8–85 years). Almost all patients (218/226, 96.4%) were submitted to total thyroidectomy with or without lymphadenectomy. The large majority of patients (172/226,76.1%) were located in Pisa, where there is a national referral center for thyroid disease. At final histology, an extrathyroidal extension was found in 51/226 patients (22.6%), while lymph node metastases were observed in 33/226 cases (14.6%). Three (1.3%) patients presented distant metastases at diagnosis. In 105 cases, the tumor was multicentric (46.5%), and it was bilateral in 103 (45.6%); the mean diameter was 1.7 ± 1.4 cm.

The clinical and pathological features of TC patients according to the five areas of Grosseto Province are reported in Table 3. Thyroid cancer patients living in the Metal Hills District and in Albegna Hills presented a slightly younger mean age at TC diagnosis compared to other TC patients living in different areas. No significant differences were observed among the other analyzed parameters (gender, histotype, variants of PTC, diameter of the tumor, multifocality and bilaterality of thyroid tumors, rate of lymph node and distant metastases, and tumor extension) between the five groups.

## 4. Discussion

Our study highlights a high incidence of thyroid cancer in Grosseto Province, located in southern Tuscany (central-western Italy) for both sexes. Our epidemiological observation does not currently have a direct demonstration of the causes underlying this increase. However, the observation that only less than 30% of cases consisted of microcarcinomas, the presence of a relatively high percentage of aggressive variants of PTC (in particular solid and tall cell variants), and a fairly large increase in the rate of bilaterality and multicentricity of the tumor make a screening effect unlikely to be the sole cause of this increased incidence. The potential role of environmental factors cannot be excluded. 

Southern Tuscany is an important mining province because of the magmatism (intrusive and effusive) occurring with the post-collisional events of the Northern Apennines orogenic belt between Neogene and Quaternario (8.5 Ma–0.3 Ma). The mineralizations in Metal Hills and Amiata Mountain, defined as the “Tuscan Magmatic Province”, are represented by ore deposits, including sulfides (especially pyrite), iron, and silver [25,26,27]. 

Mining works were mostly concentrated in a time span of about one hundred years, from the second half of the 19th century up to the 1980s, when they definitively ceased. On a global scale, the mining activity in Tuscany was very minor. Natural processes of rock weathering and the release of acid into mines led to the aqueous dispersion of toxic elements in the environment. The current remediation of mining areas is only partial and is still underway [28]. A significant transport of mercury (Hg) and arsenicum (As) is documented in the streams draining the district, and consequently, potentially toxic elements are widely distributed in the surrounding environment [29]. Mercury, arsenicum, lead, cadmium, and aluminum are among the most dangerous cancer-related toxins to which we are all exposed. As a recognized carcinogen (class 1, according to International Agency for Research on Cancer—IARC classification). Mercury, the second most toxic substance in the world, is present in the air, as well as in food, water, personal care products, and dental amalgam used as fillings. Metals are also found in less obvious sources, such as cosmetics [30,31,32,33,34].

Metals are naturally occurring elements that cannot be broken down and are not biodegradable. Heavy metals, even at low levels of exposure, can act as carcinogens, probably with a synergistic effect of a complex mixture of interacting chemicals, by causing genetic and epigenetic alterations in susceptible cells and favoring their malignant transformation [35,36,37].

The metallogenic area of Grosseto Province includes the Metal Hills District that hosts many Cu-Pb-Zn deposits, the pyrite (± barite ± Fe oxides) deposits of Gavorrano, the Hg deposits of the Monte Amiata area, and the Sb deposits of the Capalbio–Monti Romani belt. One of the most remarkable geological features is Mt. Amiata (1738 m above sea level), dominated by a homonymous volcanic system (0.3–0.2 Ma old). The Mt. Amiata area hosts the third largest Hg district in the world, overlapping with a present-day geothermal system that is exploited for energy production [38]. A significant transport of mercury is documented in the streams draining the district [39]. Recently, a detailed epidemiological study (InVETTA survey) based on the results of over 10 years of research documented the presence of arsenic, mercury, and thallium above the usual limits in water and studied a potential association with health issues. Food-related exposure to heavy metals was found in the survey population, evidenced by the increase in the levels of these metals in urinary samples. The authors, evaluating a possible association between heavy metal contamination and the risk of developing cancer, highlighted the risk only for thyroid cancer that appears to be associated with exposure to the concentrations in the air of arsenic, mercury, and H2S emitted by geothermal power plants [40].

These geological characteristics are not present in neighboring areas such as the Siena and Arezzo Provinces, where a lower incidence has been observed.

A recent paper reported an excess risk of TC incidence in patients living in Italian National Priority Contaminated Sites (NPCS) with a documented presence of endocrine disruptors (EDs), suggesting a potential etiological role of residential exposure to EDs in the development of TC in the communities in these areas [41]. The authors analyzed data from only 13 of these 59 NPCs, and it is interesting to note that 2 of these 59 are located in the province of Grosseto (Orbetello, ex-Sinoco) or in very close proximity (Piombino).

Given the particular geological composition of southern Tuscany, the intense exploitation of mineral resources, and the demonstration of widespread contamination by heavy metals, we cannot exclude a link with the increased incidence of thyroid cancer, although there is currently no demonstrative evidence to confirm this type of association. It is noteworthy that studies conducted in a different Italian region, Sicily, have shown that the inhabitants who live in particular types of environments, such as volcanic areas, have an increased risk of thyroid cancer [42]. In addition, they demonstrated that the concentration levels of many metals are significantly higher in the urine of these subjects than in the urine of residents in adjacent non-volcanic areas, suggesting the hypothesis that an excess of heavy metals could be involved in the pathogenesis of thyroid cancer [43]. Experimental studies have shown that As and Hg, in particular, were slightly increased in the thyroid but not in the other examined tissues of the patients living in volcanic areas [44].

We cannot exclude a potential role of other environmental factors such as exposure to radiation, the progressive increase in iodine intake, or not yet identified endocrine disruptors chemicals as a cause of the increased incidence of TC in our study population. Indeed, pesticides, persistent organic pollutants (POPs), endocrine-disrupting chemicals (ECDs), bisphenol A (BPA), phthalates, and polychlorinated biphenyls (PCBs) can cause cancer through various actions, including hormonal imbalance, DNA damage, causing inflammation in tissues, and turning genes on or off. For example, the Pesticides Action Network (PAN)—an international organization, also present in Italy—has tried to educate and identify alternatives to the use of harmful pesticides. Another possible risk factor in the Grosseto area is the high presence of radon which could be responsible for sick building syndrome (SBS). Indeed, a recent report by Arpat, which evaluated the presence of radon values beyond the normal limit, showed that 54% of Tuscan municipalities with higher radon presence were located in Grosseto Province [45,46,47,48]. Exposure to toxic chemicals will make it harder for the immune system to eliminate cancer [49]. However, heavy metal contamination appears to be the most likely risk factor in southern Tuscany for the demonstration of the presence of diffuse metal pollution. Both in vitro and in vivo experiments suggest that chronic exposure to heavy metals may alter the biology of stem thyroid cells, leading to a predisposition to malignant transformation [50]. Different metals may accumulate at higher levels in the thyroid than in other tissues, and this selective accumulation of one or more trace elements with a carcinogenic effect could explain the predominant increase in thyroid cancer more so than cancers of other tissues, as reported in the InVETTA study [40].

An interesting observation is that, according to the latest Globocan 2020 estimates, Cyprus has the highest incidence of thyroid cancer in Europe [10]. The name Cyprus comes from the Latin word “*cuprum*”, which means copper, and the island is naturally rich in heavy metals, as evidenced by the origin of its name. Accumulating evidence reveals that the surroundings of the abandoned copper mining site of Cyprus are contaminated, at least to some extent, by the dispersal of potentially toxic elements, an environmental situation similar to the one that occurred in southern Tuscany [51].

A limitation of our study is that we have no information about patient outcomes because about half of TC patients are followed in other regional hospitals, or about the professional activities of the TC patients. The higher frequency of TC observed in male patients might suggest that one’s line of work might have a greater effect than the type of diet on the incidence of TC. Moreover, the analysis of the presence of heavy metals in the blood and urine of TC patients was performed only in the minority of patients living in the territory of the Monte Amiata area. However, the presence of heavy metals has been amply demonstrated to be present in the soil, water, and air [27,52,53].

In conclusion, our study documents a high prevalence of cancer in a geographic area with heavy metal pollution. The aim of our next studies will be to verify the presence of heavy metals in thyroid tumor tissue in patients residing in the province of Grosseto compared with the tissue of patients living in other provinces of our region. Further studies will serve to clarify whether this is a casual association or whether there is a pathogenetic link between TC incidence and metal pollution.

Since the worldwide increase in environmental metal pollution is accompanied by a parallel increase in thyroid cancer incidence, we retain, in agreement with other authors, that the thyroid gland may act as a sensitive indicator of the possible health damage caused by environmental heavy metal pollution.

## Figures and Tables

**Figure 1 biomedicines-11-00298-f001:**
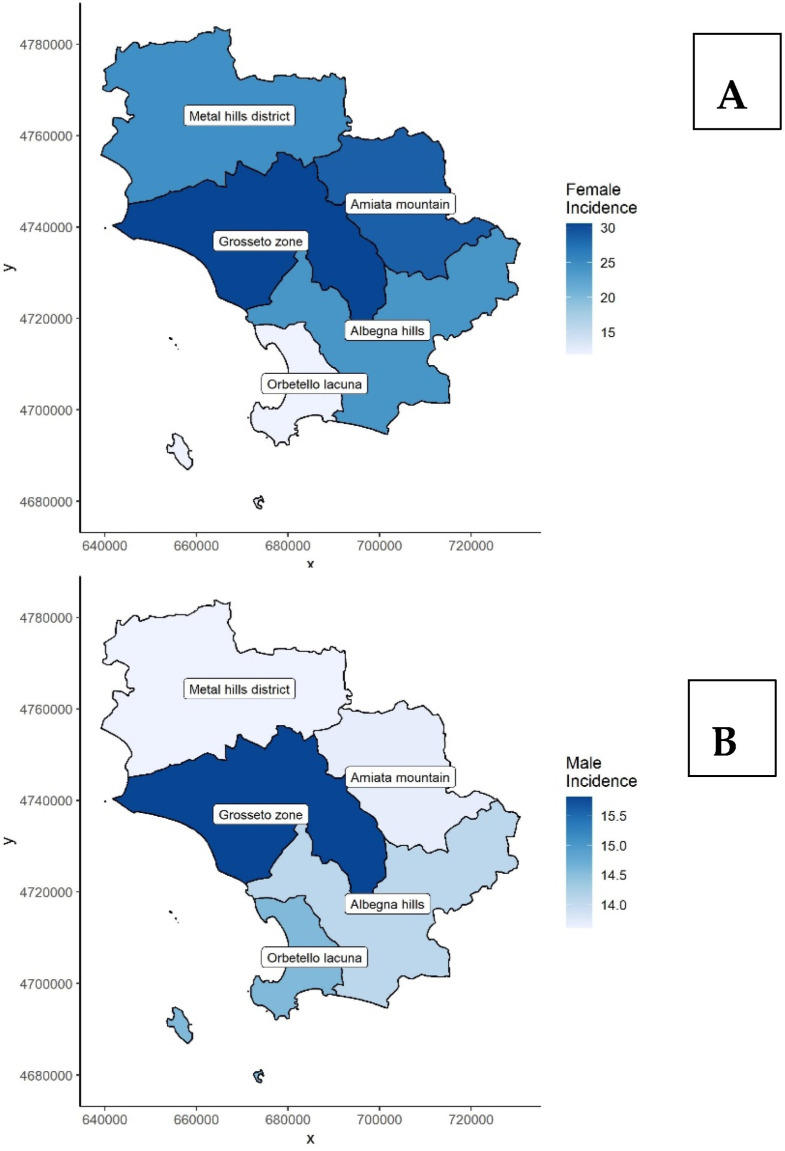
Thyroid cancer (TC) incidence rate distribution in the five areas of Grosseto province differentiated between females (panel **A**) and males (panel **B**).

**Table 1 biomedicines-11-00298-t001:** Thyroid Cancer Incidence in Southeast Tuscany * and in Italy **.

	MALES	FEMALES
**GROSSETO ***		
Cases	76	159
Crude IR (95% CI)	17.77 (10.70–27.75)	34.46 (24.41–46.72)
EU-standardized IR (95% CI)	17.66 (12.46–19.89)	34.16 (28.36–39.43)
SIR vs. Italy (95% CI)	1.52 (1.17–1.95)	0.95 (0.79–1.14)
**AREZZO ***		
Cases	62	158
Crude IR (95% CI)	9.32 (5.27–15.25)	22.41 (15.97–30.57)
EU-standardized IR (95% CI)	8.58 (6.48–10.88)	22.20 (19.05 –26.36)
SIR vs. Italy (95% CI)	0.92 (0.70–1.18)	0.79 (0.67–0.92)
SIR vs. Grosseto (95% CI)	0.53 (0.40–0.68)	0.66 (0.56–0.77)
**SIENA ***		
Cases	44	142
Crude IR (95% CI)	8.58 (4.28–15.35)	22.53 (17.83–35.42)
EU-standardized IR (95% CI)	8.51 (15.86–10.94)	25.30 (20.86–29.43)
SIR vs. Italy (95% CI)	0.84 (0.61–1.14)	0.90 (0.76–1.06)
SIR vs. Grosseto (95% CI)	0.49 (0.35–0.65)	0.75 (0.63–0.88)
**ITALY ****		
Crude IR	10.1	28.2

* data from Tuscan Cancer Registry, Institute for Cancer Research, Prevention and Clinical Network (ISPRO), Florence, Italy. ** data from the Italian Association of Cancer Registries (AIRTUM 2019).

**Table 2 biomedicines-11-00298-t002:** Pathological features of our study group (*n* = 226).

Parameters	Number of Patients
**Cancer histotypes: *n* (%)**	
Papillary	208 (92)
Follicular	8 (3.5)
Medullary	7 (3.1)
Anaplastic/Undifferentiated	3 (1.4)
**Variants of PTC *n* (%)**	
Classical	48/176 (27.3)
Follicular	63/176 (35.8)
Tall cell	35/176 (19.9)
Solid	25/176 (14.2)
Diffuse Sclerosing	5/176 (2.8)
**Type of surgery: *n* (%)**	
Total thyroidectomy	218 (96.4)
Hemithyroidectomy	8 (35)
**Location of surgery: *n* (%)**	
Grosseto	30 (13.3)
Pisa	172 (76.1)
Siena	13 (5.7)
Others	11 (4.9)
**Diameter of the tumor: (cm)**	
Mean ± SD	1.7 ± 1.4
Range	0.1–12
Median	1.4
**Tumor extension: *n* (%) **** (TNM 8th Edition).	
T1a	65 (28.7)
T1b	75 (33.2)
T2	33 (14.6)
T3	51 (22.6)
T4	2 (0.9)
**Lymph-node metastases: *n* (%)**	
Yes	33 (14.6)
**Distant metastases: *n* (%)**	
Yes	3 (1.3)
**Bilaterality: *n* (%)**	
Yes	103 (45.6)
**Multicentricity: *n* (%)**	
Yes	105 (46.5)

** refer to TNM 8th edition.

**Table 3 biomedicines-11-00298-t003:** Clinico-pathological features of TC patients (*n* = 226) according to the areas of residence (Grosseto Province).

Parameters	Amiata Mountain (*n* = 16) (%)	Grosseto Zone (*n* = 120) (%)	Albegna Hills (*n* = 21) (%)	Orbetello Lacuna (*n* = 21) (%)	Metal Hills District (*n* = 48) (%)	*p*
**Age at diagnosis (yrs)**						0.06
Mean ± SD	49.6 ± 12.9	53 ± 13.7	47 ± 19.1	55.7 ± 10.8	47.7 ± 14.4	
Range	22–70	23–84	33–74	8–76	22–85	
Median	47	52	59.5	50.5	48	
**Sex**						0.6
Males	4 (25)	34 (28.3)	6 (28.6)	9 (42.8)	17 (35.4)	
females	12 (75)	86 (71.7)	15 (71.4)	12 (57.2)	31 (64.6)	
**Histotypes: *n* (%)**						0.9
Papillary	15 (93.7)	109 (90.9)	20 (95.2)	20 (95.2)	44 (91.8)	
Follicular	1 (6.3)	5 (4.1)	0	1 (4.8)	1 (2.0)	
Medullary	0	4 (3.3)	1 (4.8)	0	2 (4.2)	
Anaplastic/Undifferentiated	0	2 (1.7)	0	0	1 (2.0)	
**Variants of PTC *n* (%)**						0.3
Classical	1/13 (7.7)	29/94 (30.8)	4/16 (25)	3/16 (18.8)	11/37 (29.7)	
Follicular	6/13 (46.1)	34/94 (36.2)	3/16 (18.8)	8/16 (50)	12/37 (32.4)	
Tall cell	3/13 (23.1)	18/94 (19.1)	7/16 (43.8)	2/16 (12.4)	5/37 (13.6)	
Solid	2/13 (15.4)	12/94 (12.8)	1/16 (6.2)	3/16 (18.8)	7/37 (18.9)	
Diffuse Sclerosing	1/13 (7.7)	1/94 (1.1)	1/16 (6.2)	0	2/37 (5.4)	
**Diameter of tumor: (cm)**						0.7
Mean ± SD	1.8 ± 1.3	1.7 ± 1.3	1.5 ± 0.8	1.4 ± 0.8	1.7 ± 1.4	
Range	0.2–5	0.1–6	0.2–3	0.2–3.5	0.1–8	
Median	1.5	1.4	1.5	1.5	1.3	
**Tumor extension: *n* (%) ***						0.5
T1a	5 (31.2)	36 (30)	5 (23.8)	5 (23.8)	14 (29.2)	
T1b	2 (12.5)	38 (31.7)	9 (42.9)	11 (52.4)	15 (31.2)	
T2	4 (25)	14 (11.7)	5 (23.8)	3 (14.3)	7 (14.6)	
T3	5 (31.3)	31 (25.8)	2 (9.5)	2 (9.5)	11 (22.9)	
T4	0	1 (0.8)	0	0	1 (2.1)	
**Lymph-node metastases: *n* (%)**						0.2
Yes	3 (18.7)	14 (11.7)	6 (28.6)	4 (19)	7 (14.6)	
No	13 (81.3)	106 (88.3)	15 (71.4)	17 (80.9)	41 (85.4)	
**Distant metastases: *n* (%)**						0.8
Yes	0	2 (1.7)	0	0	1 (2.1)	
No	16 (100)	118 (98.3)	21 (100)	21 (100)	47 (97.9)	
**Bilaterality: *n* (%)**						0.3
Yes	7 (43.7)	57 (47.5)	9 (42.8)	13 (61.9)	17 (35.4)	
No	9 (56.3)	63 (52.5)	12 (57.2)	8 (38.1)	31 (64.6)	
**Multicentricity: *n* (%)**						0.3
Yes	7 (43.7)	56 (46.7	9 (42.8)	14 (66.7)	19 (39.6)	
No	9 (56.3)	64 (53.3)	12 (57.2)	7 (33.3)	29 (60.4)	

* refer to TNM 8th edition.

## Data Availability

Data supporting reported results can be found at the Institute for Cancer Research, Prevention, and Clinical Network (ISPRO) of Florence, where epidemiologic research and surveillance activities have been developing since 1988. The authors declare that all data supporting the findings of this study are contained within the article file.

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
