# Peer review of "High Incidence of Thyroid Cancer in Southern Tuscany (Grosseto Province, Italy): Potential Role of Environmental Heavy Metal Pollution"

_biomedicines, 2023, doi:10.3390/biomedicines11020298_

Round 1

Reviewer 1 Report

The submitted study focused on regional differences in the incidence of thyroid carcinoma in Tuscany. The authors identified a higher percentage of aggressive variants of papillary thyroid carcinoma in one region (Grossetto) and speculated that exposure of the population to heavy metals from the environment could be the cause. Regional differences were seen especially for men – typically only few cases were available in this group.

In the introduction more information should be provided on the regional differences, specifically in which order of magnitude they are.

This hypothesis is not convincingly supported by experimental evidence; only reference to the excretion in volcano-rich areas has been made and metal-content of the thyroid or urine excretion of the patients was not determined.

It is further not clear if the patients in the same region during their entire lives because mobility in the 21th century is higher than it used to be decades ago. The profession of the patients is also not known. Since the differences were highest for male incidence differences in the profession are more likely than in the nutrition.

Reviewer 2 Report

Authors should mention which cancers are identified and in what numbers. After all, it is known that malignant thyroid tumors are relatively rare tumors: approx. 80% of removed focal lesions of the thyroid gland or lesions found incidentally in preparations after lobectomy (lobectomy - removal of a lobe of the thyroid gland) or thyroidectomy (thyroidectomy - removal of the entire thyroid gland) for other indications are not shows histopathological features of malignancy. Among the malignant thyroid neoplasms, there are four main histopathological forms of carcinomas: papillary, follicular, medullary and anaplastic, as well as neoplasms of other origin: thyroid lymphomas (usually B-cell) and thyroid sarcomas. The authors should mention this. In addition, the authors did not mention other factors that may be important in the development of cancer, which is very important. Factors affecting the formation of thyroid cancer include, among others: iodine deficiency in food (goiter endemic in iodine-poor areas), excessive thyroid stimulation by TSH (trophic factor for thyrocytes - primary hyperpituitarism, hypothalamic diseases), ionizing radiation (e.g. in patients undergoing radiotherapy for thymus tumors in childhood, head and neck tumors), genetic and hereditary factors: activation of ras, ret oncogenes, suppression of p53 genes, presence of growth factors and receptors for: TSH, cytokines, epidermal growth factor (EGF) ), fibroblast growth factor (FGF). The presence of metals does not have to be so important. Authors should specifically describe this in the manuscript. The work lacks information about the condition of patients. And it is known that it often happens that patients with thyroid cancer may not feel any specific symptoms directly related to the development of cancer. The first symptom of thyroid cancer may be a nodule - detected accidentally in an ultrasound examination in people under constant endocrinological control for other indications, or a painless but palpable lump in the neck in the thyroid gland. The authors should add that it should be borne in mind that many adults have small nodules, of which less than 5% are diagnosed as malignant. When the tumor grows rapidly, the neck circumference enlarges, resulting not only from the expansion of the tumor itself, later - swelling of the surrounding tissues, but also, in very advanced stages, from the enlargement of regional lymph nodes. In addition to the aforementioned, there may be pain in the front of the neck, a change in the timbre of the voice and hoarseness resulting from the mechanical pressure of the tumor on the area of the speech apparatus or nerves responsible for phonation, gradually increasing problems with swallowing or breathing, bone pain in advanced stages. All this can be of decisive importance should be described in the article. It is worth noting that although thyroid cancers are most common in patients with euthyroidism (proper functioning of the thyroid gland), there may also be symptoms of hypothyroidism (apathy, fatigue, weakness, drowsiness, difficulty concentrating, memory disorders, depression, intolerance to cold, chronic constipation, dry, cold, pale skin, subcutaneous swelling, brittle hair, sinus bradycardia, menstrual problems) and hyperthyroidism (hyperactivity, increased sweating, heat intolerance, palpitations, shortness of breath, feeling weak, weight loss despite increased appetite , shaking hands, warm and clammy skin, insomnia. This needs to be mentioned. This is very important. The authors did not write anything about the assessment of thyroid cancer staging, which should be carried out at the next stages of diagnosis and treatment, as in the case of most malignancies according to the TNM classification, taking into account the size of the primary tumor (Tumor-T; imaging, histopathological examinations), involvement of regional lymph node disease (Nodes-N) and distant metastasis (Metastatic disease-M). This is missing from the article. In addition, it is worth noting when discussing heavy metals that mercury, lead, cadmium and aluminum are among the most dangerous cancer-related toxins to which we are all exposed. Mercury – the second most toxic substance in the world – is present in the air, as well as in food, water, personal care products and dental amalgam used as fillings. Metals are also found in less obvious sources such as cosmetics. For example, a 2014 scientific study that analyzed thirty types of lipstick used by women in China found that all brands contained lead! Aluminum, on the other hand, is present in almost all antiperspirants. Authors should add more studies by other authors to discuss their results. The discussion of the results should be increased. The authors should also mention that according to the Pesticide Action Network (PAN) - an organization whose goal is to educate and identify alternatives to the use of harmful pesticides - "chemicals can cause cancer through various actions, including hormonal imbalance, DNA damage, causing inflammation in tissues and turning genes on or off. Many pesticides are known to cause cancer and (as the Panel notes) every resident of the United States comes into contact with them on a daily basis, perhaps the same is true in Italy. Authors should check it out. Interestingly, the sick building syndrome describes buildings with potentially toxic contaminants, such as mold, radon, lead paint, and formaldehyde-containing carpeting. Flooded, new and/or renovated buildings often have large amounts of these contaminants. If the inhabitants. If Italians work or live in a sick building, they should get it cleaned or consider moving as it will be harder for your immune system to get rid of the cancer if you are exposed to mold or toxic chemicals. That is also worth mentioning. I recommend that the authors add to the references an article that is important in terms of preventing cancer: Kieliszek, M., & Lipinski, B. (2018). Pathophysiological significance of protein hydrophobic interactions: An emerging hypothesis. Medical Hypotheses, 110, 15-22. Readers will be interested in this issue. It is also very important in inhibiting the development of diseases related to the thyroid gland. What are the prospects for the future? The authors did not mention anything about it. It is worth adding information for readers that there are no scientifically documented indications, for example, for chemotherapy in differentiated cancers and medullary thyroid cancer — the use of CTH (e.g. doxorubicin monotherapy) at the stage of generalization and after exhausting the possibilities of isotopic treatment is associated with a small percentage of objective responses, and therefore with a slow course of the disease, toxic treatment is often abandoned. In addition, people with disseminated and progressive disease should be included in controlled clinical trials - the use of molecularly targeted drugs (including tyrosine kinase inhibitors) may bring new treatment options for thyroid cancer. How the authors think about it. Please comment.

Round 2

Reviewer 1 Report

I thank the authors for the answers to my comments.

Reviewer 2 Report

The article can be recommended for publication.